# Influenza Neuraminidase: A Neglected Protein and Its Potential for a Better Influenza Vaccine

**DOI:** 10.3390/vaccines8030409

**Published:** 2020-07-23

**Authors:** Luca T. Giurgea, David M. Morens, Jeffery K. Taubenberger, Matthew J. Memoli

**Affiliations:** 1LID Clinical Studies Unit, Laboratory of Infectious Diseases, Division of Intramural Research, National Institute of Allergy and Infectious Diseases, National Institutes of Health, Bethesda, MD 20892, USA; memolim@niaid.nih.gov; 2National Institute of Allergy and Infectious Diseases, National Institutes of Health, Bethesda, MD 20892, USA; dmorens@niaid.nih.gov; 3Viral Pathogenesis and Evolution Section, Laboratory of Infectious Diseases, Division of Intramural Research, National Institute of Allergy and Infectious Diseases, National Institutes of Health, Bethesda, MD 20892, USA; taubenbergerj@niaid.nih.gov

**Keywords:** influenza, neuraminidase, universal influenza vaccine

## Abstract

Neuraminidase (NA) is an influenza surface protein that helps to free viruses from mucin-associated decoy receptors and to facilitate budding from infected cells. Experiments have demonstrated that anti-NA antibodies protect animals against lethal influenza challenge by numerous strains, while decreasing pulmonary viral titers, symptoms, and lung lesions. Studies in humans during the influenza A/H3N2 pandemic and in healthy volunteers challenged with influenza A/H1N1 showed that anti-NA immunity reduced symptoms, nasopharyngeal viral shedding, and infection rates. Despite the benefits of anti-NA immunity, current vaccines focus on immunity against hemagglutinin and are not standardized to NA content leading to limited and variable NA immunogenicity. Purified NA has been shown to be safe and immunogenic in humans. Supplementing current vaccines with NA may be a simple strategy to improve suboptimal effectiveness. Immunity against NA is likely to be an important component of future universal influenza vaccines.

## 1. Introduction

Seasonal influenza vaccination is the best option available to counteract the significant worldwide burden of morbidity and mortality caused by both epidemic and pandemic influenza. Global vaccination efforts have met considerable challenges in the face of the unpredictable nature of influenza evolution. The frequent reassortment and constant mutation of this RNA virus significantly impacts population immunity and plays an important role in the persistence of influenza strains, as the underlying immunity from past strains becomes less effective against mutated antigen [1]. Consequently, protective immunity against influenza is considered short lived, and updated yearly vaccination is necessary [2]. Even small changes, such as a single amino-acid mutation, can lead to antigenic drift, causing significant vaccine mismatches due to diminished antibody binding activity [3,4]. Reassortment between two or more viral strains in an infected host, known as an antigenic shift, can produce novel virus strains that evade existing immunity in large sections of the population, causing pandemics [5,6,7]. Vaccine efficacy has varied widely, from 10% to 60% [8], depending on the match of vaccine antigens to circulating strains, though other factors such as patient population characteristics and vaccine preparation may also have significant impact on vaccine efficacy [9,10,11].

In recent years, the influenza research community has unified around a principal goal, the development of a “universal influenza vaccine” (one affording significantly broader protection), which would satisfy the purpose of not only expanding protection against a plurality of influenza strains and increasing the duration of vaccine-associated immunity, but also potentially preventing pandemics and host-switched infections from poultry and mammalian influenza viruses [12]. Proposed strategies for a universal vaccine have included the induction of antibodies against multiple hemagglutinin (HA) and neuraminidase (NA) subtypes, the conserved ‘stalk’ portion of HA, and the highly conserved M2 protein, among other approaches [2,13]. NA is a critical glycoprotein present on the surface of influenza A and B viruses with enzymatic activity that facilitates successful viral budding, making it an attractive target for vaccination. Although it has long been demonstrated that anti-NA antibodies reduce influenza viral replication, transmission, and pathology in animals as well as viral shedding and clinical disease in human studies, influenza vaccine research efforts have for many decades focused on immunity against HA [14,15]. In fact, current licensed vaccines are not quantitatively standardized to NA amount or antigenicity. Consequently, both the NA content and NA immunogenicity of vaccines can be quite variable, including a lack of NA activity altogether [16,17,18]. All the available data suggest that the addition of NA antigens to current and future vaccine strategies may significantly improve vaccine efficacy, may reduce the impact of novel virus pandemics, and could be an important step forward in the quest for a more broadly protective vaccine. In this review, we discuss these data and attempt to identify further areas of research needed that could inform the development of novel vaccines that target the NA of influenza viruses.

## 2. Neuraminidase and Its Potential as a Vaccine Target

NA was first described in 1956 as an external viral protein with enzymatic activity for sialic acid [19,20,21,22]. It is a mushroom-shaped homotetramer on the viral surface, roughly 100 × 100 × 60 Å, anchored by a hydrophobic region in the stalk near the N-terminal [23]. Nine subtypes of NA associated with type A influenza viruses consistently found in birds and in some mammals form two genetically and structurally distinct groups: group one consists of N1, N4, N5, and N8 while group two consists of N2, N3, N6, N7, and N9 [24,25]. Only two of these, N1 and N2, have been associated with viruses capable of establishing endemicity in humans [2], though N3 [26], N6 [27], N7 [28], N8 [29], and N9 [30] have been implicated in epizootic outbreaks of avian influenza A. NA from influenza B forms a separate structural group while two additional NA subtypes, N10 and N11, have been identified in influenza virus-infected bats, but these proteins lack sialidase activity [31]. NA contributes to influenza pathogenicity through multiple mechanisms, though all stem from its ability to cleave the α-ketosidic linkage between the terminal sialic acid and the adjacent sugar residue (Figure 1A) [32]. NA helps nascent viral particles to bud from infected cells by cleaving the hemagglutinin receptor on the host cell, propagating infection [33]. Though NA has been demonstrated to not always be necessary for budding [34], it is not surprising that increased NA activity, such as that associated with the 2009 influenza A/H1N1 pandemic virus (pH1N1), has been linked with increased droplet transmissibility in ferrets [35]. The balance of HA and NA activity is also critically important for optimal viral propagation [35]. High HA activity can be detrimental in some cases, but viral fitness can be restored through increased NA activity [36]. NA also helps to release the virus from mucin-associated decoy receptors and impedes NK cell activity by protecting HA from NK cell binding [37,38,39]. There is also evidence that the NA mutations acquired from viral passage through MDCK cells may also permit H3N2 NA to induce hemagglutination in lieu of HA [40,41]. Furthermore, the influenza-associated stripping of terminal sialic acids on epithelial cells can facilitate the adherence of *Streptococcus pneumonia*, predisposing to secondary bacterial pneumonia, a major cause of influenza-associated mortality [42]. Higher NA activity has also been associated with increased bacterial adherence which can be attenuated by NA chemical inhibitors [42,43]. Antiviral agents that directly inhibited NA’s conserved enzymatic active site were developed in the 1990s, showing efficacy in decreasing symptom scores, duration of illness, inflammatory markers, and viral titers in human subjects [44,45,46].

There are other characteristics associated with NA which make it an attractive target for improved influenza vaccines. Two principal factors have been identified as responsible for the loss of immunity from season to season: diminishing human antibody titers and viral antigenic drift. Multiple studies have shown that both anti-NA and anti-HA antibodies wane over time, but experiments in mice have suggested anti-NA antibodies decline at slower rates compared to anti-HA antibodies [47,48]. In healthy human volunteers, high anti-NA titers were present in 83% of participants at baseline, compared to high HA inhibition (HAI titers) titers in only 38%, suggesting either that anti-NA antibodies wane more slowly than anti-HA antibodies in humans or that re-exposure to antigenically similar NA (which drifts more slowly than HA, as described below) maintains antibody titers higher than HA [49]. However, in a separate study, the volunteers vaccinated with an inactivated trivalent vaccine or lived-attenuated vaccine had gradually decreasing HAI and NA inhibition (NAI) titers for 18 months post-vaccination, with NAI titers decreasing to lower levels than HAI titers [47]. This discrepancy might be due to a difference in the immune response between natural infection and vaccines, which may have a low NA content, leading to poor immunogenicity [16]. Further studies are necessary, but improvement in the maintenance of high antibody titers may require more aggressive induction of T-cell and memory B-cell immunity, possibly through the use of adjuvants [12].

Antigenic drift (due to viral mutation in non-conserved areas of HA and NA) leads to the decreased affinity of existing antibodies, even if high antibody titers are maintained [50]. There is evidence suggesting that NA has a lower rate of antigenic change over time compared to HA [51], and that drift occurs independently in the two proteins [4,52]. A slower rate of antigenic drift in NA suggests that anti-NA antibodies could maintain an effective binding activity longer than anti-HA antibodies. Consequently, NA-based vaccines may not need to be updated as frequently as HA-based vaccines. However, it has been postulated that the slower rate of antigenic drift in NA may be a result of decreased selective immune pressure from current strategies [52,53]. Mutations accumulated in H1N1 NA from 1991 to 2006 did not lead to the loss of anti-NA antibody affinity [4]. Interestingly, in certain cases, a single point mutation is sufficient to impair the antibody inhibitory activity by antibodies against NA or HA [3,4]. Even so, an independent rate of antigenic drift suggests a multi-target approach against both HA and NA may be a viable strategy to improve vaccine efficacy.

## 3. In Vitro and Animal Studies

Early influenza investigators identified the promise of anti-NA antibodies through in vitro and in vivo experiments. Antibodies can bind to critical epitopes near the NA catalytic site, thereby impairing viral budding and escape from mucin-associated decoy receptors (Figure 1B), as well as mediating cellular cytotoxicity [54]. In early experiments, anti-NA antibodies were shown to reduce plaque size and number as well as decrease viral yield in cell culture [55]. Aggregates of viral particles were seen on electron microscopy after the treatment of influenza virus with anti-NA sera [56]. Subsequent experiments in animal models further supported the importance of anti-NA antibody immunity (Table 1). Mice, immunized with purified neuraminidase to induce anti-NA antibody, had improved outcomes in the face of influenza virus challenge, as demonstrated by decreased viral titers, decreased weight loss, and fewer lung lesions [48,57,58,59,60,61,62,63]. Similar findings were seen in chicken, ferret, and guinea pig experiments [64,65,66]. Furthermore, decreased susceptibility to transmitted infection with influenza A/H2N2 virus was seen in mice previously immunized with purified N2 NA [67], and decreased transmission of influenza B virus was demonstrated in NA-immunized guinea pigs exposed to infected guinea pigs [66]. The passive immunization of mice with anti-NA antibody also improved survival in the setting of lethal infectious challenge [68,69].

Beyond protection against a homologous virus, the cross-reactive activity of anti-NA antibodies against heterologous viral strains (strains with the same NA subtype, but who have undergone antigenic drift) has been demonstrated in numerous studies, illustrating the broadly protective aspect of anti-NA immunity. Mice immunized with purified N2 NA prior to viral challenge with a drifted influenza A/H3N2 virus had lower viral pulmonary titers compared to those immunized with H3 HA [58,74]. Similar findings were replicated with influenza B, and furthermore, cross-protection was demonstrated to correlate with the magnitude of NA titers [73]. In another experiment, antibodies against N1 generated from the exposure to pre-2009 seasonal H1N1 influenza had cross-reactivity to 2009 pH1N1 NA, correlating with reduced lethality [80]. Moreover, there is evidence supporting NA-mediated cross protection against viral strains with mismatched HA, including avian viruses. Immunization with 1931 swine H1N1-protected chickens against lethal viral challenge with influenza A/H7N1 virus [81]. Mice immunized with H1N1 NA DNA vaccines or virus-like particles (VLPs) containing NA from 2009 pH1N1 were protected from lethal influenza A/H5N1 virus challenge [82,83,84], and ferrets immunized with inactivated H1N1, purified N1 or N1-based VLPs were partially protected against lethal H5N1 influenza challenge [85,86]. Pigs infected with H1N1 demonstrated high titers of cross-reactive anti-NA antibodies and had decreased symptoms and shedding after subsequent H5N1 challenge [87]. These studies demonstrated clear cross-protection, while other experiments did show that there are limits to the degree of cross-reactivity, especially across different NA groups. For example, immunization with an H3N2 NA DNA vaccine was protective against heterologous H3N2 virus challenge, but not against H1N1 challenge [78,88]. Similarly, vaccination of mice with H5N1-VLPs or N1-VLPs was protective against H5N1 and heterologous H1N1 but not to a more distant H1N1 strain [89]. However, one experiment has shown a promise of heterosubtypic protection (protection against strains containing different NA subtypes) induced by VLPs containing both A/PR/8/34 H1N1 NA and M1. Vaccinated mice had increased survival in the face of not just A/PR/8/34 H1N1 challenge but also against heterosubtypic A/Philippines/82 H3N2 challenge [90].

To further mechanistically establish the cross-protective potential of an anti-NA antibody, the efficacy of monoclonal antibodies and passive immunization have been demonstrated. Passive antibody transfer from mice exposed to heterologous H1N1 and H5N1 led to increased survival in mice undergoing lethal H1N1 challenge [69]. Multiple studies utilizing a single monoclonal antibody directed against conserved portions of NA, demonstrated broad protection in mice against lethal challenge with H1N1, pH1N1 and H5N1 [91,92,93]. Mutations at the catalytic site in the N1 vaccine antigen, such as I365T and S366N, have been shown to expand the breadth of induced antibody cross-reactivity leading to neutralizing capabilities across groups, including against influenza A/H7N9 virus [94]. This may provide a sophisticated strategy for the induction of broadened immunity via an NA-based universal influenza vaccine. Antibodies with greater broad binding and neutralizing potential have been isolated [95]. Human monoclonal antibodies isolated from an H3N2-infected individual have been demonstrated to bind and inhibit all nine influenza A NAs and influenza B NA in vitro. Furthermore, passive immunization with a monoclonal antibody protected mice from influenza challenge with H3N2, H7N2, H6N3, H4N6, H1N7, H7N9, pH1N1, H5N1, H15N5, H10N8, recombinant H6/1 N4 (H6 globular head attached to H1 stalk), and influenza B viruses [96]. These results are promising with respect to the broad and possibly universal protective potential of anti-NA immunity.

It is important to note that NA-induced immune responses against influenza can involve mechanisms beyond antibodies. As opposed to HA-associated immunity, which can prevent influenza infection from being established, NA induces an infection-permissive immunity which may provide an opportunity for the development of complementary immunologic responses [70]. Stimulation of phagocytosis by macrophages has been shown to be related to NA-associated surface desialylation [97]. NA epitopes, presented via both MHC class one and class two pathways [98], have been shown to increase NK activity [99,100,101] and stimulate CD8+ cytotoxic T-cells [102]. The importance of viral entry into cells was demonstrated in an experiment comparing anti-HA and anti-NA immunity. Mice were vaccinated with H1 or N1, then challenged with a sublethal dose of 2009 pandemic H1N1 virus followed by a second lethal challenge with H3N2 virus. H1-vaccination prevented infection with H1N1 but also the development of a CD8+ response. N1-vaccinated mice developed infection-permissive immunity (demonstrated by low viral titers but no weight loss) which facilitated the development of a CD8+ response. Upon subsequent H3N2 challenge, H1-vaccinated mice suffered 90% lethality whereas N1-vaccinated and placebo-vaccinated mice had 100% survival, presumed by the investigators to be secondary to a cross-protective CD8+ response [103]. These findings illustrate yet another advantage of NA-based vaccines over traditional HA-focused vaccines and add to an extensive body of literature demonstrating the cross-protective potential of NA-associated immunity in in vitro and animal experiments.

## 4. Importance of NA Immunity in Humans

Significant evidence exists in humans to corroborate the findings seen in animal studies, including NA’s impact on viral shedding, NA’s impact on symptoms, and even NA’s cross-protective potential. Protective effects of anti-NA antibodies in humans were seen with the advent of the pandemic influenza A/H3N2 virus in 1968, which occurred due to reassortment between seasonal H2N2 influenza and an unidentified avian H3 virus [104]. The H3 antigen was novel but immunity to N2 was present in persons with prior exposure to circulating H2N2. Higher pre-outbreak titers of anti-N2 antibodies were associated with a decreased risk of infection [105]. Similarly, in Japanese school children, high levels of pre-outbreak anti-NA antibodies were protective and correlated better with protection compared to high anti-HA antibodies [106]. In human volunteers without anti-HA antibodies, but with varying levels of anti-NA antibodies, higher titers of anti-NA antibody were associated with the absence of symptoms, decreased nasopharyngeal viral shedding, and decreased viral shedding duration after influenza challenge [107,108]. Recent influenza challenge studies have confirmed these findings and further demonstrated that levels of anti-NA antibody in serum correlated better with reduced influenza disease severity than levels of anti-HA head and stalk antibodies, with high pre-existing anti-NA titers correlating with fewer number of symptoms, decreased symptom severity, decreased symptom duration, and decreased duration of viral shedding [49,109]. High pre-existing anti-NA antibody levels were also shown to be protective in two community-based studies (though not in a third) [110,111,112], and in vaccinated individuals the protective effect of anti-NA antibodies was shown to be independent of the protection afforded by anti-HA antibodies [113]. Furthermore, the presence of anti-NA antibodies in both serum and nasopharyngeal washes correlated with resistance to infection [110,114].

Anti-NA immunity can be induced or augmented through vaccine efforts and can be cross-protective in humans against reassorted strains. Eight hundred and seventy-five schoolchildren vaccinated with a recombinant virus containing equine HA and N2 from A/Port Chalmers/1/73 H3N2 experienced decreased seasonal H3N2 infection rates [115] and 2400 military recruits vaccinated with adjuvanted H2N2 experienced decreased infection rates and H3 seroconversion compared to controls, in the face of the H3N2 pandemic which occurred a few weeks later [116]. Infection rates during the 1968 H3N2 pandemic were inversely correlated with pre-outbreak anti-NA antibody titers, regardless of N2 origin (from vaccination with H3N2 or H2N2), demonstrating the potential of anti-NA immunity to curtail pandemics [117]. However, outside of the generous human research precipitated by the 1968 H3N2 pandemic, there has been a paucity of investigation exploring NA-specific vaccination in humans.

## 5. NA-Based Vaccine Strategies

Despite the benefits of anti-NA immunity outlined above, existing influenza vaccines have had variable NA content with some vaccines having nearly undetectable levels (Table 2) [16,113,118,119,120,121]. NA amount can vary wildly between lots of vaccines made by a single manufacturer [119]. Chiefly responsible is the absence of quantitative quality control measures in place for NA activity or amount, even though they correlate well with immunogenicity [16,17]. Furthermore, neuraminidase activity decreases over time in stored vaccine lots, which may be the result of storage conditions, especially with particular strains [16,122]. Consequently, anti-NA antibody response rates after vaccination with standard vaccines tend to vary widely [113,123]. Both intranasal and subcutaneous vaccination induce rises in anti-HA antibodies more consistently than in anti-NA antibodies [124]. Boosts in anti-NA antibodies with inactivated H3N2 or recombinant H3N2 vaccines in the early 1970s were seen in anywhere from 41% to 93% of participants [124,125]. Population characteristics, including exposure history and pre-existing anti-NA antibody titers, may be important confounding factors but response rates to modern vaccines have not improved or become more consistent. Vaccines from 2008/2009 showed a similar variation in response, with the induction of serum anti-NA antibodies against H1N1 and H3N2 ranging from 17 to 57% and from 0 to 73%, respectively [123] and inactivated vaccines from the 2013–2014 season contained between 0·02 µg and 10·5 µg of N1 across different manufacturers [120]. In a separate study, the high dose vaccine contained almost eight times as much NA as the standard dose vaccine and was found to induce more robust anti-NA antibody responses [126].

NA content in vaccines appears to correlate with anti-NA antibody response rates [16,126,128]. Increasing the NA content of vaccines and the subsequent anti-NA immunity could be a reasonable next step in improving vaccination efforts. This could be accomplished by standardizing the vaccine content of NA, as is done for HA. In the past, technical difficulties limiting the large-scale and accurate assessment of NA activity restricted efforts to address this problem [129,130,131,132], but advances in techniques such as the miniaturized thiobarbituric acid method [132], the 4-methylumbelliferyl-N-acetylneuraminic acid (MU-NANA) assay [16], the enzyme-linked lectin assay [133], and the enzyme-linked immunosorbent assay [121,134] have simplified this analysis. More recent techniques may further improve NA quantitation. For example, NA has been shown to react with TR1 releasing fluorophore in a stoichiometric fashion, allowing for the measurement of absolute amounts of NA in a fast and throughput manner [127].

Increasing the content of NA in existing vaccines may require modifications to current manufacturing practices which use an inactivated egg- or cell-grown influenza virus as a substrate for HA and NA antigen [135,136], and since NA is present in lower amounts compared to HA on the viral surface (A/Aichi/2/1968 H3N2 virions have ~50 NA spikes in clusters amid ~300 HA spikes) [137], it is inevitably present in lower amounts in the final product as well. Regardless, the variability in NA content seen across vaccines by different manufacturers suggests that the capability to optimize manufacturing processes to increase NA immunogenicity already exists [113,118,119,120]. It is important to note that vaccine storage may also impact the stability of the NA protein. Storage at 4 °C and in buffer containing calcium or magnesium optimizes NA protein stability and may provide a possible explanation for the low NA content in older vaccine lots [16,138]. Buffer optimization may potentially present a relatively easy solution to the NA deficiency problem, though more research is needed to address this question.

The direct administration of NA or supplementation of purified NA to existing vaccines are alternative strategies which have been shown to avoid the problem of antigenic competition in mice [72]. NA has been produced for vaccination purposes through chromatography purification of virus [77,79], recombinant Baculovirus-infected cell lines [58,78,120], yeast expression systems [76], and mammalian expression systems [65]. It is important to note that the choice of expression system can affect the glycosylation pattern of expressed protein, which can impact immunity [68]. Immunization with hypoglycosylated NA produced in α-1,6-mannosyltransferase defective yeast induced higher anti-NA antibody titers in mice than vaccination with glycosylated NA produced in wild type yeast [139]. Beyond live and inactivated virus vaccines, anti-NA immunity has been successfully induced in humans with hybrid virus [140], VLPs [141], and chromatography-purified NA [79]. Purified NA given to 88 human subjects was well tolerated, and at high dose, increased antibody titers in 80% of the subjects compared to 55% in those who received the 1994–1995 inactivated trivalent vaccine [79]. Furthermore, Phase one, two and three studies looking at the clinical efficacy of NA-based vaccines in humans, either given individually or in combination with existing influenza vaccines, are desperately needed.

Antigenic competition between HA and NA may be an additional factor responsible for poor NA responses to both vaccines and live infection, despite the sufficient NA antigen present, as seen in both animal models and human studies. Vaccinated subjects who already had anti-HA antibody against vaccine strains were less likely to develop responses against NA [118,142]. Anti-NA antibody development was lower than anti-HA antibody in primed adults, while unprimed children had higher response rates to NA than HA [118]. This phenomenon overlaps with the concept of original antigenic sin: the exposure to influenza in non-naïve individuals recalls the memory B cell responses that have affinity against antigen from an initial influenza exposure [143]. Mice primed by infection with H3N1, H3N2 and H3N7 followed by vaccination with H7N2 had a greater anti-NA response compared to the primed mice given H3N2 vaccination. Anti-NA response was reciprocal to anti-H3 response [144]. Further experiments confirmed the analogous findings with N1. Humans vaccinated with reassorted H7N1 (containing N1 from H1N1) before the H1N1 challenge had a more robust NA response compared to the humans vaccinated with homologous H1N1 before the H1N1 challenge [145]. However, the effects of antigenic competition were avoided through the separation of the two antigens, particularly with the use of purified NA and HA protein, which induced robust antibody responses in mice to both [63,70,74]. In a study looking at influenza B, increased levels of anti-NA antibodies were seen in mice given recombinant NA and HA compared to the mice given a live or inactivated vaccine [73]. Therefore, the administration of purified NA may not only supplement the lack of NA antigen in many marketed influenza vaccines to improve immunogenicity, but may possibly provide an elegant solution to the problem of antigenic competition between HA and NA.

## 6. Conclusions

While humanity waits for the next inevitable influenza pandemic, seasonal influenza continues to be responsible for significant morbidity and mortality across the planet. A rapid mutation rate and the presence of a large reservoir of diverse viruses in animal species have provided influenza with a robust capability to adapt and evade immunity induced by natural infection and vaccination. Existing vaccines, which have focused on HA-associated immunity, have provided suboptimal protection against seasonal strains, even when well matched. NA-based immunity may have the potential to address many of the deficiencies associated with current vaccines. Antibody titers against NA have been shown to be a more robust correlate of protection than anti-HA antibodies [49]. NA has demonstrated decreased and independent antigenic drift to HA. Anti-NA antibodies have also shown cross-reactive potential against drifted strains and avian viruses. One anti-NA antibody has shown binding capacity to all nine avian influenza A NA subtypes, leading to protection in mice against viral challenge with 15 heterosubtypic strains [96]. Since NA-content is not quantified in existing inactivated influenza vaccines, changing existing manufacturing practices to optimize and standardize NA-content and stability may be a rapid way to improve existing vaccine strategies [16]. Alternatively, NA can be supplemented or administered separately, and this may evade problems with antigenic competition [63,70]. Further study of NA supplementation to current vaccines should be pursued while the development of broadly protective, universal influenza vaccines incorporating an NA-based component should be moved rapidly through pre-clinical and clinical trials as it is likely that NA-induced immunity will be an important piece of future influenza vaccine strategies.

## Figures and Tables

**Figure 1 vaccines-08-00409-f001:**
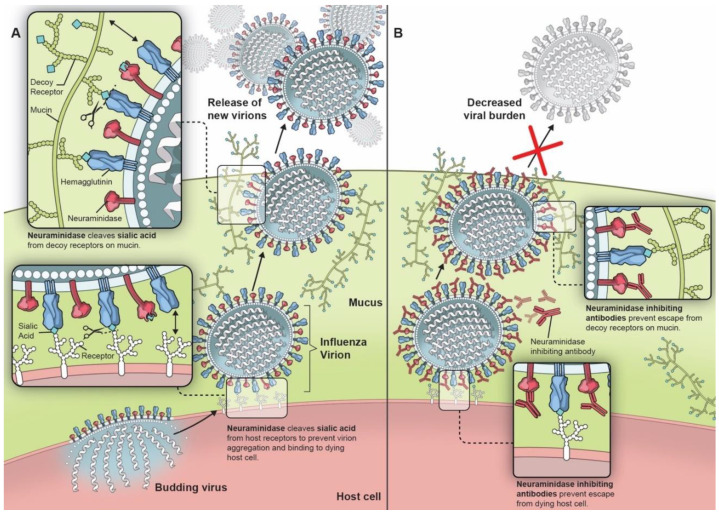
Importance of neuraminidase on influenza viral lifecycle (**A**) and the effect of neuraminidase inhibiting antibodies (**B**).

**Table 1 vaccines-08-00409-t001:** Neuraminidase (NA) vaccine outcomes by species and production methods.

Subject	Neuraminidase Manufacturing Strategy	Outcome in NA-Vaccinated Subjects	Reference
White leghorn chickens	Electrophoresis-purified influenza A N2	Increased NAI titers, decreased tracheal and cloacal viral titers	[64]
Guinea pigs	Baculovirus expression system influenza B NA	NAI titers, increased ELISA antibody titers, decreased nasal wash virus titers, decreased transmission	[66]
Manor Farm (MF-1) mice	Electrophoresis-purified influenza A N2	Increased NAI titers, decreased pulmonary virus titers, diminished lung lesions	[57]
BALB/c mice	Chromatography-purified influenza A N2	Increased NAI titers, increased ELISA antibody titers, decreased weight loss, decreased pulmonary virus titers with homotypic and heterotypic challenge	[48,59,60,61,62,63,70]
BALB/c mice	Baculovirus expression system influenza A N1, N2 and influenza B NA	Increased NAI titers, increased ELISA antibody titers, decreased pulmonary virus titers with homotypic and heterotypic challenge	[58,68,71,72,73,74,75]
BALB/c mice	Yeast expression system influenza A N2	Increased survival	[76]
New Zealand rabbits	Chromatography-purified influenza A N2	Increased NAI titers	[77,78]
Ferrets	Human embryonic kidney cell expression system influenza A N1	Increased NAI titers, decreased pulmonary virus titers, decreased lung pathology	[65]
Humans	Chromatography-purified influenza A N2	Increased NAI titers, increased ELISA antibody titers	[79]

NAI = neuraminidase inhibition. ELISA = enzyme-linked immunosorbent assay.

**Table 2 vaccines-08-00409-t002:** NA activity and concentration of influenza vaccine preparations.

Vaccine Type	Vaccine Year	NA Activity (mU/mL)	NA Concentration (μg/mL)	Reference
Monovalent whole virus (H3N2)	1968/1969	112,000 ^a^	43	[118]
Monovalent split virus (B)	1973/1974	21,000 ^a^	165	[118]
Bivalent whole virus (H3N2 + B)	1973/1974	78,000 ^a^	284	[118]
Bivalent whole virus (H3N2 + B)	1974/1975	164,000–184,000 ^a^	372–692	[118]
Trivalent whole virus	1975/1976	206,000 ^a^	596	[118]
Trivalent split virus	1975/1976	50,000 ^a^	135	[118]
Bivalent whole virus (H3N2 + H1N1)	1976/1977	10,400-60,000 ^a^	81–242	[118]
Monovalent whole virus (H1N1)	1976/1977	<500 ^a^	45–98	[118]
Monovalent (pH1N1) ^c^	2009	-	0.73–5.28	[119]
Monovalent (pH1N1) ^c^	2009	2–56 ^b^	9	[16]
Trivalent split virus	2008/2009	194–3293 ^b^	-	[16]
Trivalent ^c^	2011/2012	2–3105 ^b^	22	[16]
Trivalent ^c^	2012/2013	4521 ^b^	-	[16]
Trivalent subunit (egg derived)	2013/2014	-	5	[120]
Trivalent subunit (cell derived)	2013/2014	-	0.02	[120]
Trivalent split virus (egg derived)	2013/2014	-	10.5	[120]
Trivalent split virus (egg derived)	2013/2014	-	4.4	[120]
Quadrivalent split virus (egg derived)	2015/2016	-	2.7	[127]
Quadrivalent split virus (egg derived)	2015/2016	-	3.9	[127]
Quadrivalent split virus high-dose (egg derived)	2015/2016	-	12.9	[127]
Trivalent split virus (egg derived)	2015/2016	-	2.4	[127]
Trivalent subunit (egg derived)	2015/2016	-	3.4	[127]
Quadrivalent live-attenuated virus	2015/2016	-	1.1	[127]
Quadrivalent split virus (egg derived)	2016/2017	-	2.8	[127]
Quadrivalent split virus (egg derived)	2016/2017	-	3.5	[127]
Quadrivalent split virus high-dose (egg derived)	2016/2017	-	9.5	[127]
Trivalent split virus (egg derived)	2016/2017	-	1.6	[127]
Trivalent subunit (egg derived)	2016/2017	-	2.8	[127]
Quadrivalent live-attenuated virus	2016/2017	-	0.4	[127]
Quadrivalent split virus (egg derived)	2017/2018	-	2.0	[127]
Quadrivalent split virus (egg derived)	2017/2018	-	3.2	[127]
Quadrivalent split virus high-dose (egg derived)	2017/2018	-	7.9	[127]
Trivalent split virus (egg derived)	2017/2018	-	1.5	[127]
Trivalent subunit (egg derived)	2017/2018	-	3.1	[127]
Quadrivalent live-attenuated virus	2017/2018	-	0.7	[127]

^a^ Activity measured using the periodate–thiobarbituric acid method. ^b^ Activity measured using 4-methylumbelliferyl-N-acetylneuraminic acid (MU-NANA) assay. ^c^ Manufacturing details not available.

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
