# Peer review of "Influenza Neuraminidase: A Neglected Protein and Its Potential for a Better Influenza Vaccine"

_vaccines, 2020, doi:10.3390/vaccines8030409_

Round 1

Reviewer 1 Report

The manuscript entitled " Influenza Neuraminidase: A Neglected Protein and its Potential for a Better Influenza Vaccine" presented by Giurgea and colleagues is a constructive and well-argued review about NA as one important target for improving the efficacy of influenza vaccine.

Recommandations and suggestions:

Line 52: The authors claim that "NA is an indispensable glycoprotein present on the surface of influenza A and B viruses with enzymatic activity necessary for successful viral budding" but some reports mentioned the successful replication of influenza viruses lacking NA segment. Although this is a very rare published event, the subject could be raised in the manuscript.

Line 71: The authors referred to a review from GM Air and WG Laver (reference 23) dating 1989 for the NA characterization. They should replace it with the latest recent review from GM Air (Air. (2012) Influenza neuraminidase. Influenza and Other Respiratory Viruses 6(4), 245–256.

Line79: Several recent publications described the concept of HA/NA balance and explain that both proteins must have an affinity, and the NA a catalytic activity, which must be compatible between them for the emergence of a virus with a good fitness and transmissibility. Could the authors mention this point with an appropriate reference.

Lines 139 Table 1: The authors could include the data references for each line of the Table in an additional column.

Line 251 Table 2: The authors could include the data references for each line of the Table in an additional column and put the table on a single page for improving the reading.

Line 263: Few additional words about the principle of TR1 titration would be appreciated.

Line 267: The authors could be more precise about the amount of HA and NA on the viral surface (see review from GM Air 2012).

Minor comment: 

Line 39 : "causing cause" could you verify please.

Author Response

Line 52: The authors claim that "NA is an indispensable glycoprotein present on the surface of influenza A and B viruses with enzymatic activity necessary for successful viral budding" but some reports mentioned the successful replication of influenza viruses lacking NA segment. Although this is a very rare published event, the subject could be raised in the manuscript.

  • Line 58-59: “indispensable” changed to “critical”, “necessary for” changed to “that facilitates”. Line 85: added “Though NA has been demonstrated to not always be necessary for budding,”. Added reference 34 (Liu, 1995) to support claim.

Line 71: The authors referred to a review from GM Air and WG Laver (reference 23) dating 1989 for the NA characterization. They should replace it with the latest recent review from GM Air (Air. (2012) Influenza neuraminidase. Influenza and Other Respiratory Viruses 6(4), 245–256.

  • Line 77: 1989 review by Air replaced by 2012 review by Air

Line79: Several recent publications described the concept of HA/NA balance and explain that both proteins must have an affinity, and the NA a catalytic activity, which must be compatible between them for the emergence of a virus with a good fitness and transmissibility. Could the authors mention this point with an appropriate reference.

  • Line 88-90: added “The balance of HA and NA activity is also critically important for optimal viral propagation.” Added reference 35 (Yen, 2011). Also added: “High HA activity can be detrimental in some cases, but viral fitness can be restored through increased NA activity.” Added reference 36 (Wagner, 2000).

Lines 139 Table 1: The authors could include the data references for each line of the Table in an additional column.

  • Line 158: Table 1, added another column with references

Line 251 Table 2: The authors could include the data references for each line of the Table in an additional column and put the table on a single page for improving the reading.

  • Line 278: Table 2, added another column with references. Page-break added to put table all one page but will defer to journal staff for final formatting. Corrected row 17 to (egg-derived) from high-dose. Also added 18 more lines of data from the TR1 paper below.

Line 263: Few additional words about the principle of TR1 titration would be appreciated.

  • Line 291-293: Added more information regarding TR1 titration, “For example, NA has been shown to react with TR1 releasing fluorophore in a stoichiometric fashion, allowing for measurement of absolute amounts of NA in a fast and throughput manner”.

Line 267: The authors could be more precise about the amount of HA and NA on the viral surface (see review from GM Air 2012).

  • Line 300: added extra information regarding exact HA and NA amounts, “(A/Aichi/2/1968 H3N2 virions have ~50 NA spikes in clusters amid ~300 HA spikes)”. Also added reference 137 (Harris, 2006).

Minor comment: 

Line 39 : "causing cause" could you verify please.

  • Line 38: Deleted duplicate “cause”.

Reviewer 2 Report

This review addresses the issue of neuraminidase as a possible antigenic stimulus capable of constituting effective immunity against the influenza virus.

The efforts made by the scientific world mainly lie in creating vaccines using hemagglutinins. Since these efforts do not result in effective immunity, emphasizing the creation of neuraminidase vaccines could be important in order to create effective and universal vaccines.

The review is interesting and well organized. Also, the English writing is really good and clear. The figure and the summary tables are interesting.

Please find below my suggestions:

Why MD near the authors?

L39: Please check, causing cause…it is correct?

The section on the Importance of NA Immunity in Humans could be improved by indicating some data on the numbers of people that have been treated with the different types of vaccines. Maybe the authors could add this info in the table 2.

The literature used for the manuscript sometimes seems a little bit old according the numerous work published in the few refers on NA based vaccines. Maybe the authors should check the last manuscripts on this topic since many papers have been published in order to enriched their work with recent data.

Author Response

Why MD near the authors?

  • MD titles removed from author section per journal style.

L39: Please check, causing cause…it is correct?

  • Line 38: Deleted duplicate “cause”.

The section on the Importance of NA Immunity in Humans could be improved by indicating some data on the numbers of people that have been treated with the different types of vaccines. Maybe the authors could add this info in the table 2.

  • Added information on number of human subjects given vaccines to boost NA immunity in lines 247, 249, and lines 319. Further information on number of subjects who have received the vaccines for which NA content is presented in table 2 is not available, to our knowledge.

The literature used for the manuscript sometimes seems a little bit old according the numerous work published in the few refers on NA based vaccines. Maybe the authors should check the last manuscripts on this topic since many papers have been published in order to enriched their work with recent data.

  • Added more recent literature to support claims in study including lines 173-174 (reference 86, Smith, 2017), line 242 (reference 112, Petrie, 2017), line 283 (reference 128, Fritz, 2012), table 1/line 179/line 312 (reference 78, Lu, 2014).

Reviewer 3 Report

Vaccines-862669 (Review): Influenza Neuraminidase: A Neglected Protein and its Potential for a Better Influenza Vaccine.

In this manuscript, Giurgea et al. discuss the reasoning of having more exploration of influenza vaccines be directed towards the neuraminidase protein (NA), since most vaccination efforts look at the HA protein. The discussion includes antibody protection and disease severity from animal and human data, including protection from mismatched viruses and drifted viruses. The manuscript provides a nice summary of what is known about antibody protection directed against the NA protein, including amounts and activity units towards the NA of many past vaccines. This manuscript fills a much-needed void in the literature. While the function of the NA protein is well-known in the influenza field, its vaccine potential is not. The manuscript is well-written and my comments are minor.

Specific Comments:

Lines 84-85: Please add the appropriate reference(s) to this sentence.

Tables 1&2: Citations should be included in the table (in addition to what the authors have already provided in the text).

Reference 53/Line 456: There is a special character instead of a letter in this reference.

Author Response

Lines 84-85: Please add the appropriate reference(s) to this sentence.

  • Lines 93-95: We rephrased and combined the unreferenced sentence with the following sentence for improved clarity and included the appropriate reference 42 (McCullers, 2003).

Tables 1&2: Citations should be included in the table (in addition to what the authors have already provided in the text).

  • Citations added to both tables.

Reference 53/Line 456: There is a special character instead of a letter in this reference.

  • Special character replaced.

Reviewer 4 Report

The authors present a nice review examining the interesting question of how Influenza neuraminidase (NA) could be a potential good target to improve Influenza vaccine. They extensively described both historical and future potential benefits that could be brought by focusing on NA-based vaccines. Overall, this review is very informative and helpful. I include some minor items to address below.

  1. Page 1, Line 18: "Experiments demonstrated THAT anti-NA antibodies……”
  2. Page 1, Line 19: "challenge with numerous strains while BY decreasing pulmonary……”
  3. Page 1, Line 21: "challenged with influenza A/H1N showed ELICITED THROUGH anti-NA immunity……”
  4. Page 1, Line 39: Authors can remove the word “cause” and keep “causing”
  5. Page 1, Line 39: “mismatch” has to be plural with “mismatches”
  6. Page 2, Line 60: “suggests” should be plural “suggest” because of “All available data”
  7. Page 3, Line 113: "There is evidence suggesting THAT NA has lower ……”
  8. Page 5, Line 167: “has” should be plural “have” because of “monoclonal antibodies and passive imunization”
  9. Page 5, Line 175: "Antibodies with increasingly GREATER broad binding……”
  10. Page 5, Line 188: "MHC class one and class two mechanisms PATHWAYS, have been shown……”
  11. Page 6, Line 219: "anti-NA antibodies was shown TO BE independent of the protection……”

Author Response

  1. Page 1, Line 18: "Experiments demonstrated THAT anti-NA antibodies……”
  • Page 1, line 17: correction made
  1. Page 1, Line 19: "challenge with numerous strains while BY decreasing pulmonary……”
  • Page 1, line 17: correction made
  1. Page 1, Line 21: "challenged with influenza A/H1N showed ELICITED THROUGH anti-NA immunity……”
  • Page 1, line 20: correction made, but in a different way than recommended: “challenged with influenza A/H1N1 showed THAT anti-NA immunity…”
  1. Page 1, Line 39: Authors can remove the word “cause” and keep “causing”
  • Page 1, line 38: correction made
  1. Page 1, Line 39: “mismatch” has to be plural with “mismatches”
  • Page 1, line 38: correction made
  1. Page 2, Line 60: “suggests” should be plural “suggest” because of “All available data”
  • Page 2, line 66: correction made
  1. Page 3, Line 113: "There is evidence suggesting THAT NA has lower ……”
  • Page 3, line 132: correction made
  1. Page 5, Line 167: “has” should be plural “have” because of “monoclonal antibodies and passive imunization”
  • Page 5, line 186: correction made
  1. Page 5, Line 175: "Antibodies with increasingly GREATER broad binding……”
  • Page 5, line 194: correction made
  1. Page 5, Line 188: "MHC class one and class two mechanisms PATHWAYS, have been shown……”
  • Page 5, line 207: correction made
  1. Page 6, Line 219: "anti-NA antibodies was shown TO BE independent of the protection……”
  • Page 6, line 243: correction made